# Aerosols and Clouds data processing and optical properties retrieval algorithms for the spaceborne ACDL/DQ-1

Guangyao Dai[1], Songhua Wu[1,2,3], Wenrui Long[1], Jiqiao Liu[4], Yuan Xie[4], Kangwen Sun[1], Fanqian Meng[1], Xiaoquan Song[1,2], Zhongwei Huang[5], Weibiao Chen[4]

[1]College of Marine Technology, Faculty of Information Science and Engineering, Ocean University of China, Qingdao, 266100, China
[2]Laoshan Laboratory, Qingdao, 266200, China
[3]Institute for Advanced Ocean Study, Ocean University of China, Qingdao, 266100, China
[4]Key Laboratory of Space Laser Communication and Detection Technology, Shanghai Institute of Optics and Fine Mechanics, Chinese Academy of Sciences, Shanghai 201800, China
[5]Key Laboratory for Semi-Arid Climate Change of the Ministry of Education, College of Atmospheric Sciences, Lanzhou University, Lanzhou, 730000, China

*Correspondence to*: Songhua Wu (wush@ouc.edu.cn)

**Abstract.** The new-generation atmospheric environment monitoring satellite DQ-1, launched successfully in April 2022, carries the Aerosol and Carbon Detection Lidar (ACDL) which is capable of globally profiling the aerosols and clouds optical properties with high accuracy. The ACDL/DQ-1 is a high-spectral-resolution lidar (HSRL) that separates molecular backscatter signals using an iodine filter, and has 532nm polarization detection and dual wavelength detection at 532nm and 1064nm, which can be utilized to derive aerosol optical properties. The methods have been specifically developed for the data processing and optical properties retrieval according to the specific characteristics of the ACDL system and are introduced in detail in this paper. Considering the different signal characteristics and different background noise behaviours of each channel during daytime and nighttime, the procedures of data pre-processing, denoising process and quality control are applied to the original measurement signals. The aerosol and cloud optical properties products of the ACDL/DQ-1, including total depolarization ratio, backscatter coefficient, extinction coefficient, lidar ratio and colour ratio can be calculated by the retrieval algorithms presented in this paper. Two measurement cases with use of the ACDL/DQ-1 on 27[th] June 2022 and the global averaged aerosol optical depth (AOD) from 1[st] June to 4[th] August 2022 are provided and analyzed, demonstrating the measurement capability of the ACDL/DQ-1.

## 1 Introduction

Aerosols and clouds greatly affect the Earth's climate through their direct and indirect impact on the radiation budget. The altitude of the cloud layers and their multi-layer structures can influence the radiative balance of the earth–atmosphere system by reflecting sunlight and absorbing & emitting thermal radiation. The aerosol radiation forcing significantly affects the atmospheric physical and chemical processes (Boucher et al. 2013). Therefore, as one of the largest uncertainties in global climate model, aerosols and clouds bring inaccuracy in estimating the climate change and weather forecasting (IPCC 2014).

Several studies show that local aerosols events also affect global climate change on a larger scale (IPCC 2021; Dai et al. 2022). The satellite-based observations have great potential to acquire the global aerosols and clouds information. As a well-developed active remote sensing tool, lidars can provide aerosols and clouds profile measurements with high accuracy and high spatiotemporal resolution (King et al. 1999; Winker et al. 2007). The Cloud-Aerosol Lidar and Infrared Pathfinder Satellite Observation (CALIPSO) satellite was launched in April 2006, with a primary payload of the Cloud-Aerosol Lidar with Orthogonal Polarization (CALIOP) (Hunt et al. 2009). CALIOP produces a dataset of global vertically resolved cloud and aerosol properties which gives inspiring insights to understanding the role of aerosols and clouds in the climate system (Winker et al. 2009). However, it is difficult to measure the lidar ratio with CALIOP as the Mie scattering and Rayleigh scattering are combined in the backscatter signal (Sayer et al. 2012). Lidar ratio is defined as the ratio of the aerosol extinction coefficient to the backscattering coefficient and is closely related to the physical and optical properties of the particles. The CALIOP team has developed a Hybrid Extinction Retrieval Algorithm (HERA) which retrieves both the particulate backscatter and extinction profiles from attenuated backscatter profile by including the scene classification (Young et al. 2009) as a-priori. By improving the algorithms for aerosol classification and lidar ratio selection, the accuracy of the inversion can be improved to some extent (Young et al. 2018). The previous validation studies have shown that a relatively large uncertainty would appear in extinction coefficient retrievals of aerosols and clouds as the lidar ratio is selected or modelled (Schuster et al. 2012; Balmes et al. 2019).

Taking advantage of the difference in spectral broadening, high-spectral-resolution lidar (HSRL) can separate the aerosol contribution from the molecular backscatter with a narrow bandwidth optical filter (Fiocco and DeWolf, 1968;Shimizu et al., 1983). Thus, without assuming the lidar ratio, the aerosol backscatter and extinction coefficients can be obtained simultaneously. HSRL uses several techniques to achieve a clear separation between Mie and Rayleigh scattering spectra, including the Fabry-Pérot interferometer edge technique approach (Garnier and Chanin, 1992; Flesia and Korb, 1999), interferometric fringe imaging techniques (Matthew and James, 1998) and atomic or molecular filter discrimination (She et al., 1992; Liu et al., 1997). Many HSRL instruments have been successfully implemented to measure the atmospheric parameters or particle optical properties by means of ground-based lidars and airborne lidars (Esselborn et al. 2008; Li et al. 2008; Müller et al. 2014). The Aeolus satellite, which carries a Fabry-Pérot interferometer based wind lidar (Atmospheric LAser Doppler Instrument, ALADIN), was successfully launched in 2018. It could provide the atmospheric optical properties by independently measuring the particle extinction coefficients, co-polarized particle backscatter coefficients and the co-polarized lidar ratio by the standard correct algorithm (SCA) (Flament et al. 2008; Flament et al. 2021). Additionally Aeolus optimizes the aerosol optical properties by using a maximum likelihood estimation, due to the noise sensitivity of the SCA as an algebraic inversion scheme (Ehlers et al. 2022). The Earth Clouds, Aerosols and Radiation Explorer (EarthCARE), which deploys a HSRL Atmospheric Lidar (ATLID) (Illingworth et al. 2015, Eisinger et al., 2023, Wandinger et al., 2023), is scheduled to be launched in 2024 (Wehr et al., 2023).

The Chinese first atmospheric environment monitoring satellite DQ-1 has been successfully launched on 16[th] April 2022. As an integrated detection scientific research satellite, it will serve as an important part of Chinese atmospheric environment

monitoring system. The DQ-1 is operated in a sun-synchronous orbit at the altitude of 705 km and provides global comprehensive monitoring of atmospheric particles, carbon dioxide ($CO_2$), aerosols and clouds. For the combination of the active and passive measurements, the DQ-1 is equipped with five sensors including an Aerosol and Carbon Detection Lidar (ACDL), a Particulate Observing Scanning Polarimeter (POSP), a Directional Polarization Camera (DPC), an Environmental trace gas Monitoring Instrument (EMI) and a Wide Swath Imaging system (WSI). The primary payload carried by the DQ-1 is the ACDL, which is a lidar system consisting of two different modules. The first is the aerosol-measurement module which provides aerosols and clouds profile measurements with high accuracy globally, and the second is the $CO_2$-measurement module for atmospheric column $CO_2$ observations (Chen et al., 2023). In previous studies, the ACDL scientific team measured the transmittance of the different absorption lines under different temperatures of the iodine wall and finger (Dong et al. 2018). The reliability of the space-borne system was confirmed through simulation and performance evaluation (Liu et al. 2019; Yu et al. 2018), and the impact of errors was also assessed (Zhang et al. 2018). Finally, the algorithmic and theoretical foundation for the spaceborne HSRL loaded on DQ-1 is established. The airborne ACDL prototype (A2P) was developed and mounted on an airplane to conduct the calibration and validation experiments over Qinhuangdao, China, in March 2019. The spatial and temporal developments of the atmospheric boundary layer (ABL) and aerosol distribution along the flight routes were obtained in this experiment (Wang et al. 2020; Ke et al. 2022).

In this paper, the ACDL Aerosol module (ACDL-A) measurement principles are introduced. The retrieval methods applied for the aerosol optical properties calculation are presented. Finally, two long-term global aerosol optical properties distribution measurement cases are provided. In this work, a data pre-processing process is designed that fits the characteristics of ACDL laser transmitting, receiving and data acquisition. The data quality control and segmentation denoising algorithms used in data processing show considerable noise suppression capabilities.

The paper is organized as follows. Section 2 describes the composition and working principle of the ACDL-A. Section 3 and section 4 contain the detailed description of the data processing and the optical properties retrieval algorithms. In section 5 we provide two measurement cases of optical parameter products and two months of global observations in order to demonstrate the capabilities of the presented algorithms. Conclusions and prospects are summarized in section 6.

## 2. Overview of the ACDL-A

The ACDL consists of a transmitter for generating the laser pulses down through the atmosphere, a receiver for collecting the backscatter lights from the atmospheric particles, a detection and data acquisition that measures signal strength and prepares for data downlink. The functional description of ACDL is provided in Fig 1. The transmitter contains a three-wavelength laser which emits laser beams at the wavelengths of 532 nm, 1064 nm and 1572 nm. The output pulse energies at each wavelength are monitored by energy meters. With the repetition frequency of 20 Hz, the laser emits two pulses successively with a time interval of 200 μs. This configuration of the laser emission strategy, which is called dual-pulse, has a horizontal resolution of

337 m on the surface for the original profile. Because of the dual-pulse design, we present a practical data process which is described in detail in section 3.

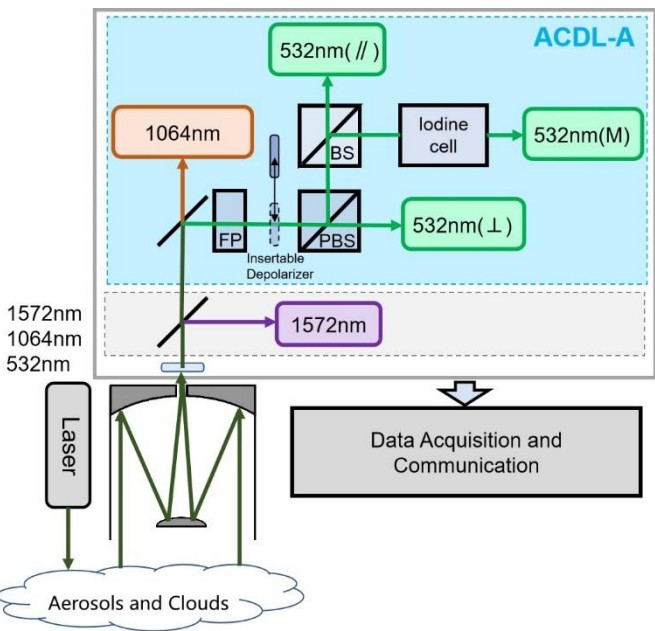


Figure 1: Schematic diagram of ACDL-A.

The ACDL takes advantage of the low transmittance valley of the iodine vapor absorption filter at 1110 line to block Mie scattering with a narrow spread. Meanwhile, the broader Rayleigh scattering can pass through (Liu et al., 1997). As shown in Fig. 1, the backscatter lights are divided into 532 nm channels (including cross-polarized and parallel-polarized channels and

high-spectral-resolution channel) and 1064 nm channel in the spectroscopic module. The cross-polarized and parallel-polarized channels at 532 nm are designed for the measurements of the linear particle depolarization ratio. After reflected by the polarization beam splitter (PBS), the lights are split again by a beam splitter (BS) and a portion of the parallel-polarized signal passes through the iodine absorption filter, thus constituting the molecular channel to estimate the optical properties of aerosol particles and clouds. Additionally, the 1064 nm channel adopts the same sampling frequency as 532 nm channels for

atmospheric detection in the ACDL-A.

Table 1: Parameters of the ACDL instrument

| Parameters | Value |
|---|---|
| Wavelength | 532.024 nm; 1064.490 nm |
| Pulse Energy | ~130 mJ@532 nm; ~180 mJ@1064 nm |
| Laser frequency stability | <2 MHz (RMS) |
| Laser divergence Angle | ⩽60 μrad@532/1064 nm; |
| Gain | 59.46@parallel; 53.4573@vertical; 32@HSRL |
| Telescope diameter | 1.0 m |
| Lidar Off-Nadir Angle | 2° |
| Laser Repetition Frequency | 20 Hz @ dual-pulse |

| Sampling rate | 50 MHz |
|---|---|
| Vertical Resolution (raw data) | 3 m@<7.5 km; 24 m (8 bin average) @>7.5 km |
| Horizontal Resolution (raw data) | ~ 330 m |

To facilitate the distinction of the parameters of each channel, the 1064 nm channel is denoted with subscript "1064", while the 532 nm channels are not specifically marked with the wavelength subscript. The superscripts of $\parallel$, $\perp$, and $M$ are used to

represent the parallel-polarized channel, cross-polarized channel and high-spectral-resolution channel at 532 nm, respectively. The signal intensity of these four channels in ACDL-A module can be expressed using the lidar equations as follows:

$$P^{\parallel}(z,\lambda) = E \frac{C^{\parallel}K^{\parallel}}{(z-z_0)^2} [\beta_m^{\parallel}(z,\lambda) + \beta_a^{\parallel}(z,\lambda)]T^2(z,\lambda) , \qquad (1)$$

$$P^{\perp}(z,\lambda) = E \frac{C^{\perp}K^{\perp}}{(z-z_0)^2} [\beta_m^{\perp}(z,\lambda) + \beta_a^{\perp}(z,\lambda)]T^2(z,\lambda) , \qquad (2)$$

$$P^{M}(z,\lambda) = E \frac{C^{M}K^{M}}{(z-z_0)^2} [f_m\beta_m^{M}(z,\lambda) + f_a\beta_a^{M}(z,\lambda)]T^2(z,\lambda) \ \text{and} \qquad (3)$$

$$P(z,\lambda_{1064}) = E_{1064} \frac{C_{1064}K_{1064}}{(z-z_0)^2} [\beta_m(z,\lambda_{1064}) + \beta_a(z,\lambda_{1064})]T^2(z,\lambda_{1064}) , \qquad (4)$$

where $P(z,\lambda)$ is the measured power at the height of $z$ received by the channel with the wavelength of $\lambda$, and $E$ is the single pulse energy, $C$ and $K$ represent the calibration constants and the system constants containing optical efficiency and gain coefficients. It is to be pointed out that in the data processing work in this paper, all heights are standardised to the orthometric height, where $z$ is the altitude to the local geodetic level. $(z - z_0)$ is the distance between the altitude where the detected

particle is located and the satellite altitude. $\beta$ denotes the backscatter coefficient at $z$ with a certain wavelength and uses the subscripts $m$ and $a$ represent the molecule and particle (aerosols and clouds). And $T^2$ is the round-trip transmittance between observation area $z$ and $z_0$. The transmittance of iodine filter for molecular scattering is denoted by $f_m$, which are function of height due to its dependence on atmospheric temperature and pressure. And the transmittance of iodine filter for aerosol scattering is denoted by $f_a$.

The molecular backscattering coefficient $\beta_m$ and the molecular extinction coefficient $\alpha_m$ can be calculated according to the Rayleigh scattering theory (Bodhaine et al. 1999; Collis and Russell 1976), and the inversion of the aerosol optical parameters can be achieved by combining the above equations (Hair et al., 2008; Liu et al., 2013).

## 3. ACDL-A data preparation

As shown in Fig. 2, the ACDL-A data processing procedure consists of data preparation, pre-processing and optical properties

retrieval. This section describes the data preparation of removing the background noise from the original voltage signal returned through the sub-channel and correcting its elevation. Additionally, the auxiliary data prepared for the inversion algorithm are also introduced in this section.

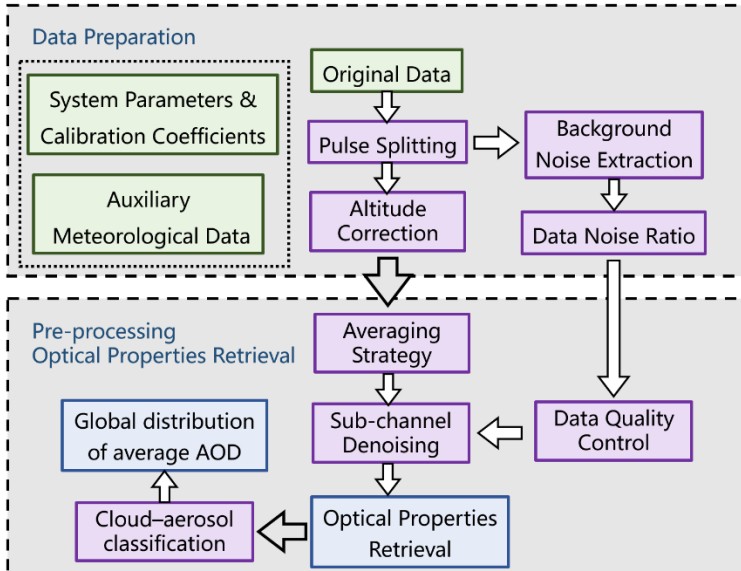

**Figure 2: Flowchart of the ACDL-A data process. The green box represents the input data, the purple box shows the data processing step, and the blue part indicates the output data products.**

### 3.1 Dual-pulse data processing

The laser in the ACDL-A transmitter applies a unique dual-pulse emission configuration, which emits a pair of pulses continuously with a repetition frequency of 20 Hz. In other words, the laser emits a total of 40 pulses in one second, where every pair of pulses are grouped together and the first pulse (odd-pulse) is launched 200 μs before the second pulse (even-pulse). As shown in Fig. 3, after triggering a set of dual-pulse, the blue line at 16.1 μs is the duration of odd-pulse emission and the orange line is the even-pulse emission duration. The time between the emission of odd-pulse (the blue line) and the start of signal acquisition (the blue dashed line) is referred to as "TimeDelay". After completing the full height data acquisition of the odd-pulse, the backscatter signals stimulated by the even-pulse are acquired immediately afterwards. Since all the channels of ACDL-A are sampled with a frequency of 50 MHz in the data acquisition module, the corresponding fundamental vertical spatial resolution is 3 m. To improve the satellite data downlink transfer efficiency, as shown in Fig. 3, the original data is 8-bin averaged at high altitude.

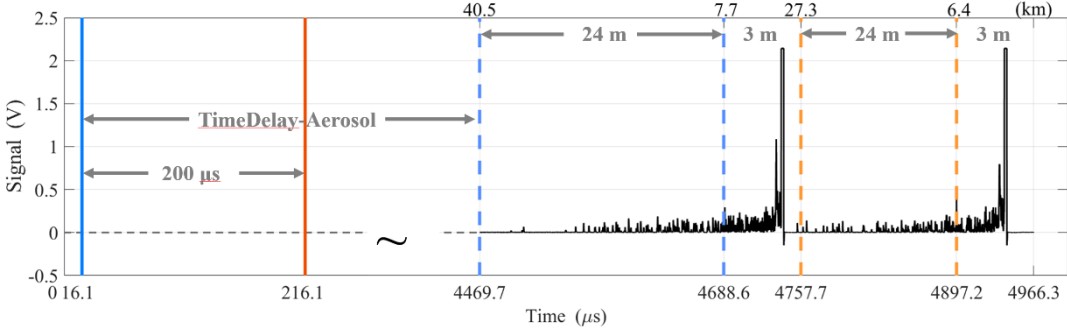

**Figure 3: Timing diagram for dual-pulse emission and acquisition.**

Since the original signal contains two pulses with different height resolution data, it is necessary to match odd-pulse and even-pulse in altitude separately in the data preparation phase. Based on the time of emission and acquisition, the position of each data point relative to the satellite can be calculated. The latitude and longitude of the laser footprint points corresponding to each set of dual-pulse were determined using spacecraft attitude and ephemeris data. The ellipsoidal heights corresponding to each data point of the odd-pulse and even-pulse are calculated separately by the WGS-84 (Lohmar 1988) coordinate system, then converted to orthometric height using the geoid height. For certain vertical resolution requirements in subsequent processing, the odd and even pulses will be averaged bin by bin by matching them to the appropriate height interval.

## 3.2 Sub-channel background denoising

In previous studies, the background noise was usually removed by subtracting the mean values of the "signals" at high-altitude during the data processing of satellite-based Lidars (Luthcke et al. 2021; Kar et al. 2018). The high-altitude "signals" are selected because than only the background noise due to the solar and dark-counts from detectors are included in the "signals". Due to the special data acquisition strategy of the dual-pulse configuration, the maximum observation altitude of the odd-pulse can be higher than 40 km (maximum detection altitude is jointed determined by the latitude and elevation). As shown in Fig. 3, after completing the signal acquisition of odd-pulse, the maximum observation altitude collected for the even-pulse is about 27 km. Hence the signal of odd-pulse is selected for background noise extraction in each channel. To avoid the effect of possible aerosol layers at high-altitude, the minimum value of the segmented-averaged signal in parallel-polarized channel and cross-polarized channel of 532 nm are selected as background noise, while the signals at high altitude are removed as background noise in the high-spectral-resolution channel and 1064 nm channel.

The amplitude of background signal in each channel is closely related to the observed location and time period. Figure 4 (a1) to (a4) and Figure 4 (b1) to (b4) show the background signal extracted from each channel during nighttime and daytime, respectively. The background signal of the 532 nm parallel-polarization channel is relatively stable and varies from 0.002 V to 0.005 V during nighttime. During daytime, due to the different irradiance at different locations and underlying surfaces, the background signal fluctuates at the range of 0.002 V to 0.5 V. The background signal of cross-polarized channel and high-spectral-resolution channel manifest similar characteristics. The cross-polarized channel background signal level at nighttime is more stable at around 0.005 V while it is more sensitive and even reaches to 1.5 V at daytime. The high-spectral-resolution channel background signal is normally below 0.002 V at nighttime and is increased obviously during the daytime. The aerosol backscatter signal from the 1064 nm channel is slightly affected by sunlight. The fluctuations in the value of the daytime background signal are related to the intensity of solar radiation at different latitudes, solar energetic events, feature type, and cloud albedo. The high values of the initial sets of nighttime observations in Fig. 4 (a1) to (a3) may result from the remaining solar radiation at dusk.

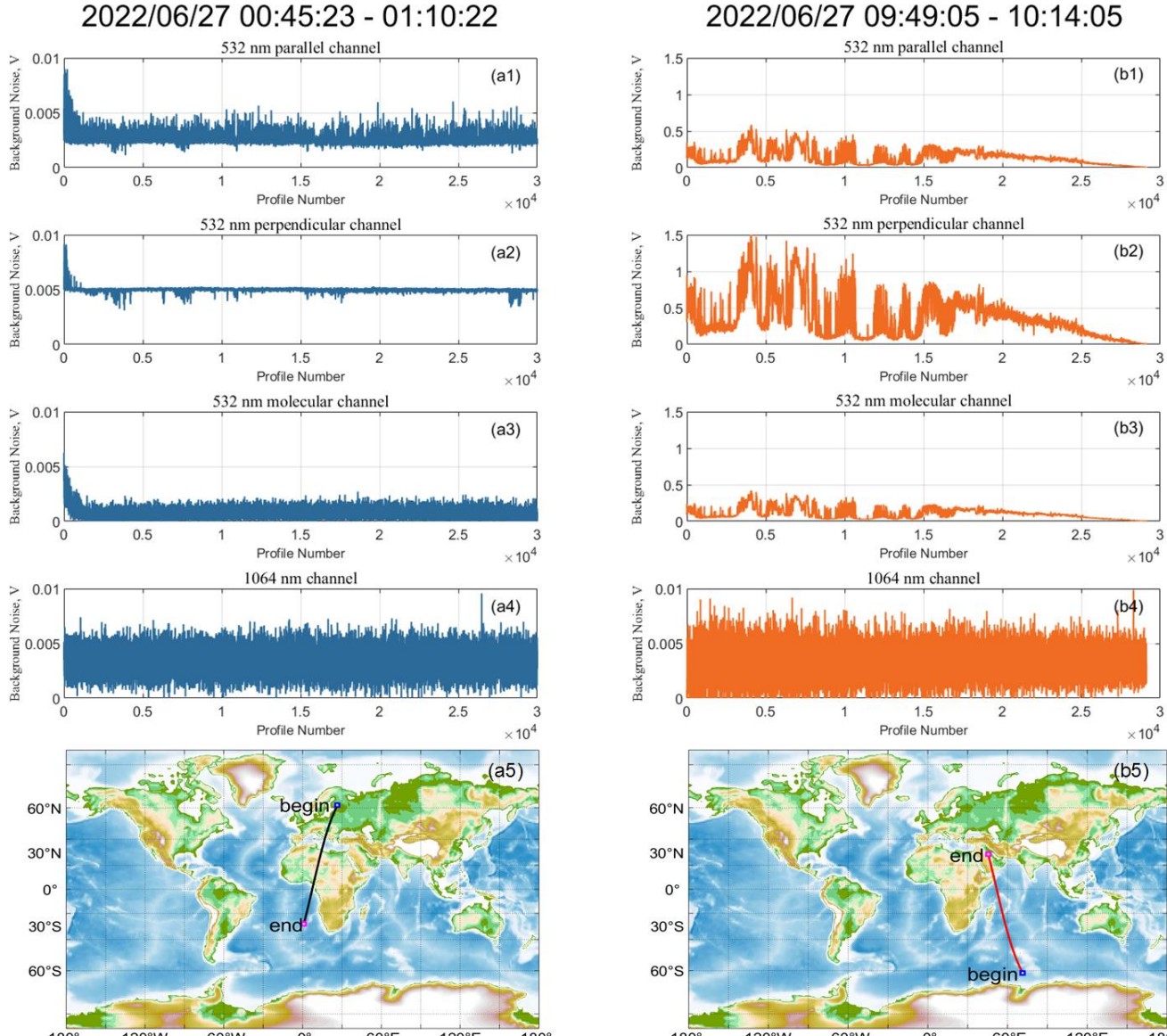

185 **Figure 4: The background signal value subtracted from each echo signal in nighttime (blue dash on the left) or daytime (orange dash on the right) segment of data, both on June 27, 2022. Corresponding to each channel in order from top to bottom, (a1, b1) parallel-polarized channel at 532 nm, (a2, b2) cross-polarized channel at 532 nm, (a3, b3) molecular channel at 532 nm, (a4, b4) 1064 nm channel, and (a5, b5) satellite orbit.**

## 3.3 Auxiliary datasets preparation

190     The auxiliary datasets contain the system parameters and meteorological data. The system parameters $C$ and $K$ for each channel are provided by the ACDL scientific team which will be not discussed in this paper. The main meteorological datasets required in the inversion algorithms include the atmospheric temperature and pressure. The calculation of the transmittance of

iodine filter for molecular scattering $f_m$ is highly reliant on them. The global temperature, ozone mass mixing ratio and pressure data are collected from the European Centre for Medium-range Weather Forecasts (ECMWF) fifth-generation reanalysis (ERA5) dataset (Hersbach et al., 2020). The ERA5 data provides hourly averaged global atmospheric parameters with a 0.25° latitude × 0.25° longitude resolution grid with 37 pressure levels. The ERA5 global data are matched according to the latitude and longitude of the ACDL-A, and the pressure gradient data is then matched to the corresponding altitude of the lidar signal using the cubic spline interpolation.

## 4. Retrieval algorithms

### 4.1 Pre-processing

#### 4.1.1 Average strategy and quality control

To improve the signal quality and reduce the interference of transient clutter, the raw data in each channel are averaged vertically and horizontally. After the assessments of different averaging scales, a vertical averaging scale of 50 m and horizontal averaging scale of 3.3 km (10 pairs of dual-pulses) for nighttime data are chosen in this work. Considering that the solar background radiation at daytime brings lower signal-to-noise ratio and more noise clutter, a wider range of averaging scale is applied during daytime. The daytime measurement data from high-spectral-resolution channel are averaged with the vertical scale of 50 m and horizontal scale of 10 km (30 pairs of dual-pulses). Figure 5 shows that after implementing the averaging strategy described above, the optical power signals for each channel are calculated using the system parameters and monthly calibration coefficients.

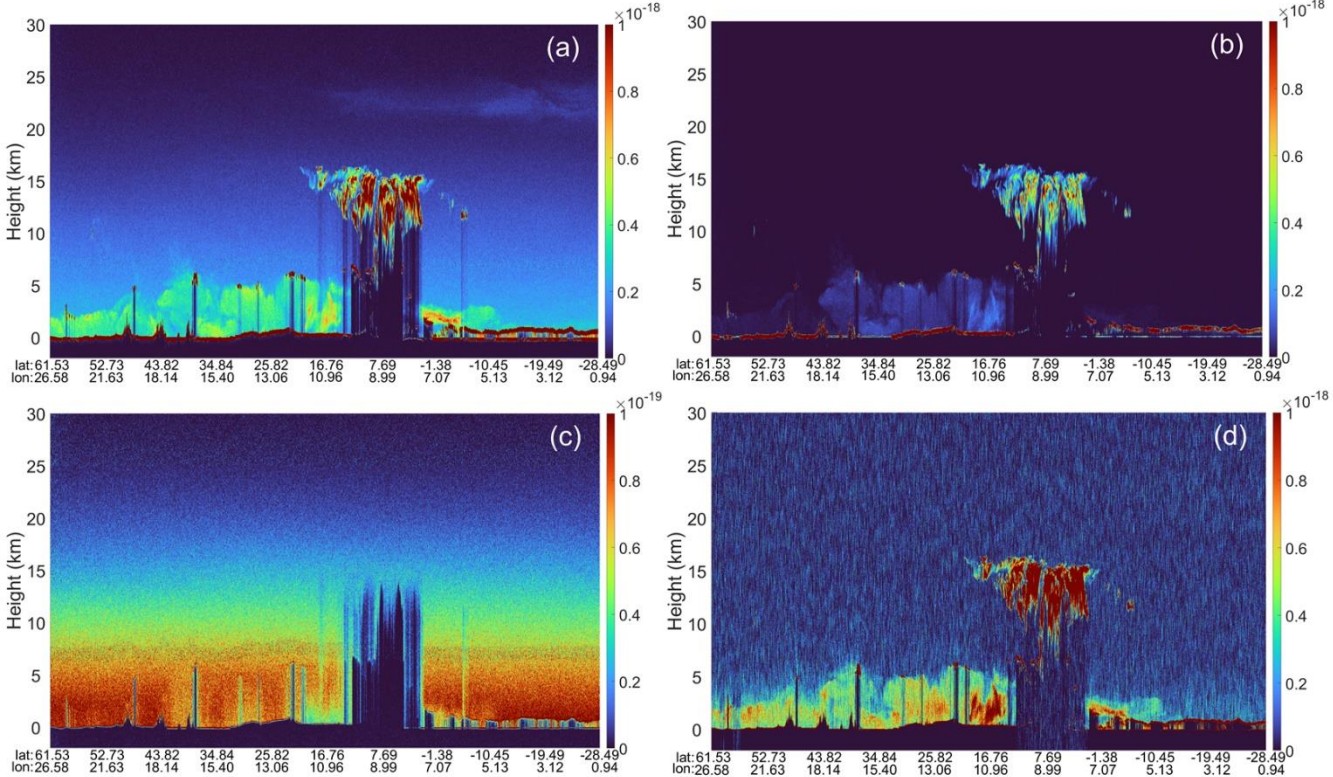

**Figure 5: ACDL-A optical power signals in (a) 532 nm parallel polarized channel, (b) 532 nm perpendicular polarized channel, (c) 532 nm molecular channel and (d) 1064 nm channel on 00:45:23 UTC to 01:10:22 UTC on June 27, 2022.**

From Fig. 5, the cloud layers at about 6-16 km and aerosol layers within the atmospheric boundary layer are observed in the parallel-polarized channel and cross-polarized channel after pre-processing procedure. However, the high-altitude aerosol layer at about 25 km is only captured by the parallel-polarized channel with a faint signal, showing weak depolarization characteristics. The molecular channel has an order of magnitude lower signal intensity than the other channels, making it more susceptible to noise, as shown in the Fig 5 (c), which shows the presence of scattering noise. Hence it is more crucial to denoise the high-spectral-resolution channel signals in the subsequent retrieval algorithms. The 1064 nm band almost has no response to molecular backscattering, and its optical power signal with low aerosols and clouds content behaves as a weakly fluctuating noise.

To evaluate the single channel echo signal quality, a parameter named Data-to-Noise Ratio ($DNR$) is introduced to calculate the ratio of the signal and subtracted background noise $P_{noise}$ at a certain height. The $DNR$ is calculated by Equation (5):

$$DNR(z,\lambda) = \frac{P(z,\lambda)+P_{noise}}{P_{noise}} .$$ (5)

The *DNR* can be used for the data quality control by setting reasonable thresholds for daytime-nighttime scenarios for each

channel. It will serve as the segmentation basis for the high-spectral-resolution channel denoising algorithm in the following subsection.

### 4.1.2 Sub-channel denoising algorithm

After completing the signal averaging process described in the previous sections, the noise still prevents the inversion of the optical parameters for the HSRL method. Due to the different optical path designs of channels in the ACDL, the signal

characteristics behave distinctly. In this work, a set of sub-channels denoising algorithms is designed especially for each channel as shown in Fig. 6. The target of the denoising algorithm is to preserve the useful information as much as possible from the optical signal data, to remove the high frequency noise and outliers, and to obtain smoother data profiles.

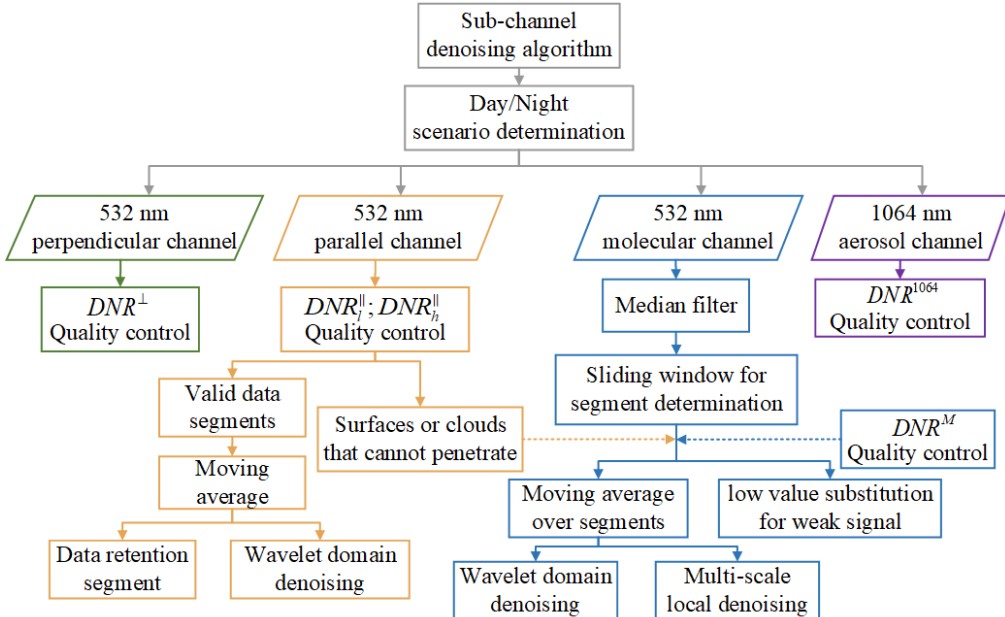

**Figure 6: Flowchart of the ACDL-A sub-channel denoising algorithms.**

After the data averaging procedure, different thresholds are chosen for each channel based on the day-night flag in the auxiliary data segment of ACDL-A. Distinct aerosol and cloud layers can be observed in the 532 nm cross-polarized channel and 1064 nm channel echo signals. The desired signal can be extracted from the overall echo profiles using a suitable threshold in the corresponding $DNR^\perp$ and $DNR^{1064}$ profiles. The parallel-polarized channel signal is divided into three parts using $DNR^\parallel$ thresholds. The valid data segments are subjected to a four-point sliding average, and the aerosol and cloud layers with high

data quality are extracted as 'Data retention segment' using thresholds, preserving as much of the original signal variations as possible. The remaining valid data segments of this profile then undergo "sym4" wavelet denoising (Daubechies, 1988; Fang and Huang, 2004), as shown in Fig 6. The echo signal is rapidly saturated at thick clouds that cannot be penetrated or at Earth's

surface, and that height position (such as the purple dashed line in Fig. 7) is localized in the corresponding high-spectral-resolution channel signal. To ensure spatial continuity of the observed profiles, a 2-D median filter is applied to the molecular channel signals of the entire orbit using a 5 × 3 window (250m vertically × 6.6km horizontally). Next, it extracts low-quality signals in the high-altitude, subsurface, and totally attenuated regions based on the DNR thresholds and corresponding parallel channel segmentation position, using a 20-point sliding window. The sliding window method ensures that occasional spikes in the signal do not interrupt the continuity of data segmentation. To maintain the continuity of subsequent data processing, low-quality signals are not eliminated but uniformly labelled as low values. The valid detection segments of the molecular channel are subject to a maximum of 20 points of slide averaging, depending on their length. Wavelet denoising and multiscale local 1-D polynomial transform are used for further denoising, with the overall goal of obtaining a smoother profile that can represent the trend of the signal.

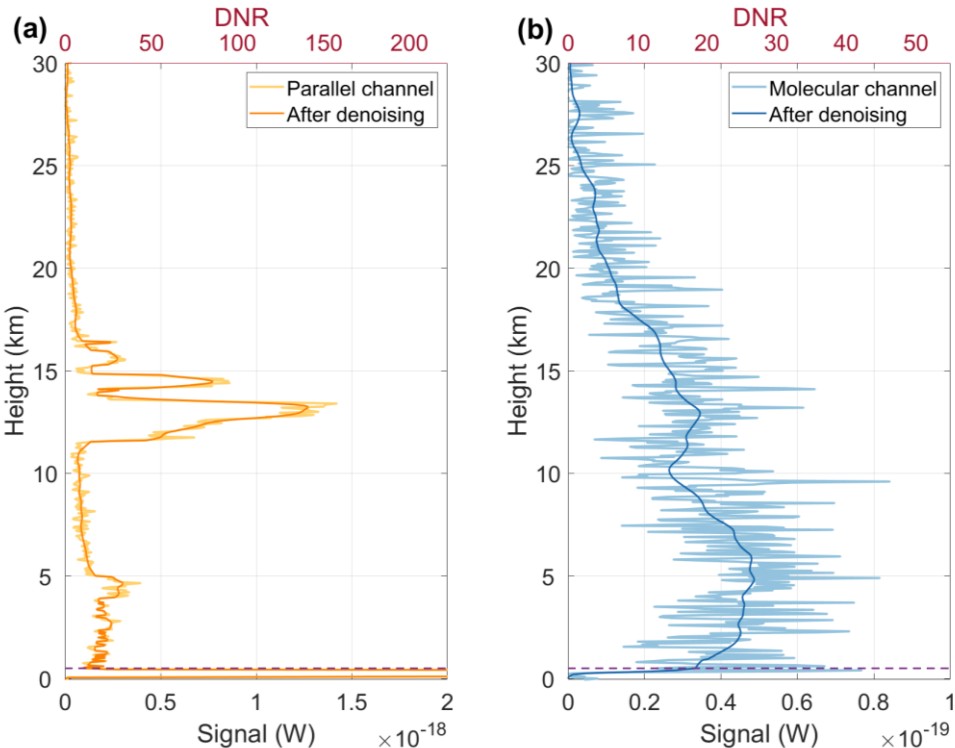

**Figure 7: 532 nm parallel polarized channel (a) and molecular channel (b) denoised profiles, the light color line is the original signal, the dark color line is after denoising.**

The signals of each channel after segmentation and denoising are re-spliced according to the height in a way that maintains local continuity. After the data pre-processing has been completed, the *DNR* is still used in subsequent inversion algorithms to represent the quality of the data.

### 4.1.3 Optical Properties Retrieval

The aerosols and clouds optical properties products of ACDL/DQ-1 include total depolarization ratio, backscatter coefficient, extinction coefficient, lidar ratio and color ratio.

The total depolarization ratio $\delta(z)$ is the ratio of total backscattering coefficient in the vertical polarization direction to that in the parallel polarization direction, which is used to describe the shape characteristics of the aerosol particles and is related to the degree of regularity of the particle shape. The total depolarization ratio can be calculated via:

$$\delta(z) = \frac{\beta_m^\perp(z) + \beta_a^\perp(z)}{\beta_m^\parallel(z) + \beta_a^\parallel(z)} = \frac{P^\perp(z,\lambda) C^\parallel K^\parallel}{P^\parallel(z,\lambda) C^\perp K^\perp} . \tag{6}$$

With the high-spectral-resolution channel, the ACDL-A enables direct detection of aerosol backscattering coefficient $\beta_a(z)$ and extinction coefficient $\alpha_a(z)$ by associating equations (1-3), which are expressed as:

$$\beta_a(z) = \beta_m(z) \frac{[1+\delta(z)][f_m(z) - f_a]\mathcal{R}(z)}{(1+\delta_m)[1 - f_a\mathcal{R}(z)]} - \beta_m(z) \text{ and} \tag{7}$$

$$\alpha_a(z) = \frac{\partial \tau(z)}{\partial z} - \alpha_m(z) . \tag{8}$$

The temperature and pressure profiles modelled with ERA5 are used to obtain the molecular scattering spectra at each height through the atmospheric model (Tenti et al. 1974; Shneider et al., 2004; Gu et al., 2013), and to calculate the $f_m(z)$ and $\alpha_m(z)$. The $\delta_m$ represents the molecular depolarization ratio, which can be also affected by the interference filter specifications and is usually set as a constant. The $\mathcal{R}(z)$ is the backscattering ratio of 532 nm parallel channel signal to the high-spectral-resolution channel signal.

The atmospheric aerosol optical depth (AOD) is defined as the integral of the aerosol extinction coefficient and molecular extinction coefficient over all heights. Conversely the gradient of AOD over height can be used to calculate $\alpha_a$. And the $\tau(z)$ can be calculated from the transmittance as in Equation (9):

$$\tau(z) = -\frac{1}{2} ln[T^2(z,\lambda)] . \tag{9}$$

The multiple scattering of clouds is considered in this algorithm, where division by the multiple scattering factor η represents
the required correction of the measured transmittance (Hu 2007; Garnier et al. 2015).

The magnitude of aerosol lidar ratio $S_a(z)$ is influenced by the aerosol absorption and scattering characteristics and is one important parameter to distinguish the aerosol types. As shown in Equation (10), it is calculated as the ratio of the aerosol extinction coefficient to the backscattering coefficient:

$$S_a(z) = \frac{\alpha_a(z)}{\beta_a(z)} . \tag{10}$$

The ACDL-A has a dual-wavelength observation capability of obtaining the attenuated color ratio $\chi'(z)$, which is defined as the ratio of the attenuated backscatter coefficient $B_\lambda(z)$ at 1064 nm to that at 532 nm, where $B_\lambda(z)$ has been calibrated by the ozone and molecular two-transmittance through the atmosphere (Vaughan et al. 2019):

$$\chi'(z) = \frac{B_{1064}(z)}{B_{532}(z)} \ . \tag{1}$$

In this algorithm, the gradient calculation on AOD profiles amplifies the noise effect from high-spectral-resolution channel.
Although the sub-channel denoising algorithm has been deployed to remove the noises, the uncertainties in the low-quality signal remain.

## 5. Results and discussion

The size of each segment is limited when ACDL-A data is transmitted back through the satellite transceiver system, and each track data is divided into two views by day and night, and each view contains two segments for subsequent data processing.
The following Fig. 8 presents the aerosol optical properties of two segments of ACDL observations at mid-low latitudes on 27th June, 2022.

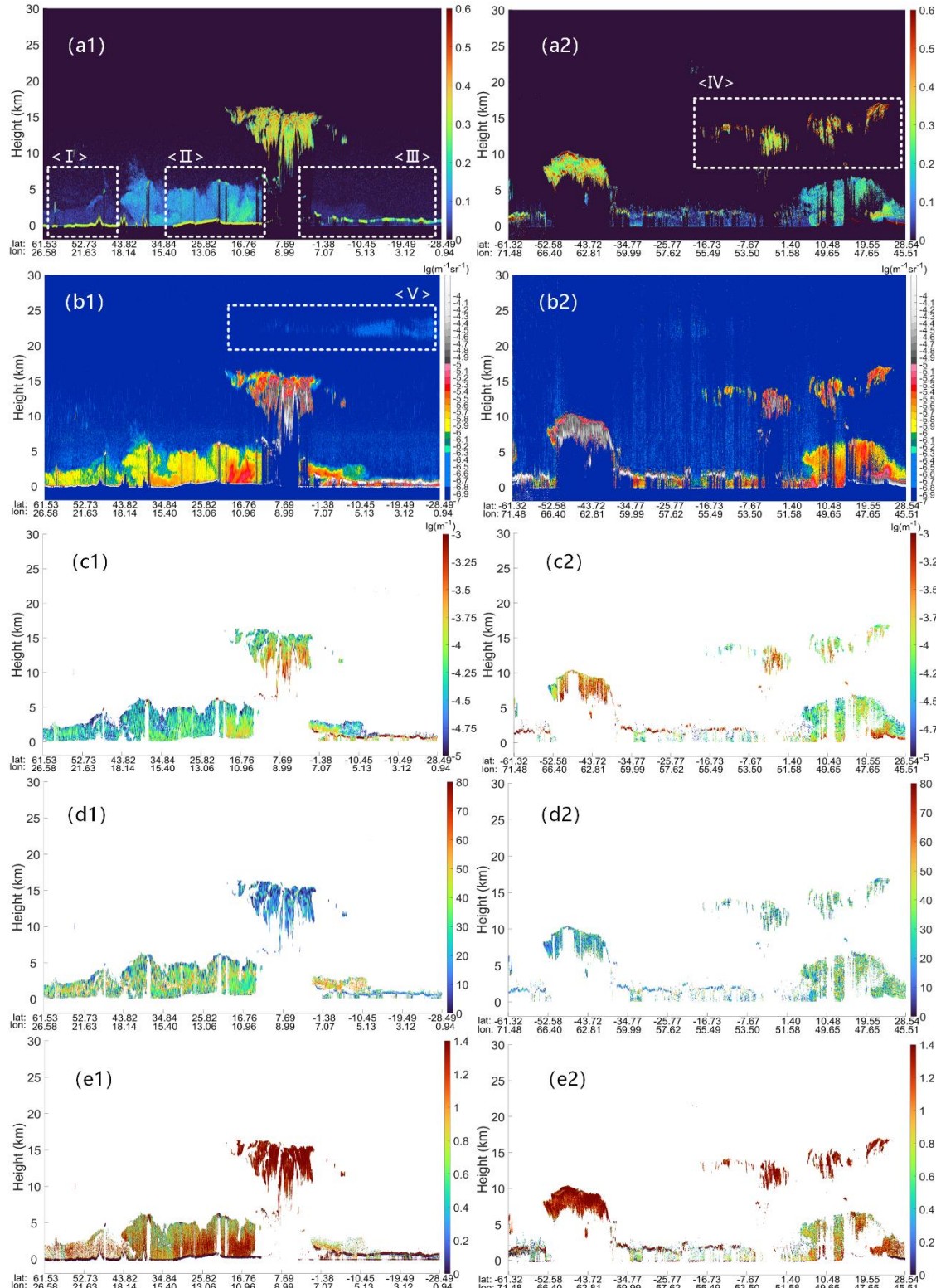

**Figure 8: Retrievals from ACDL-A measurements between 00:45:23 and 01:10:22 (nighttime, a1 to e1) and between 09:49:05 and 10:14:05 (daytime, a2 to e2) on June 27, 2022. From top to bottom: The time series of (a1, a2) total depolarization ratio, (b1, b2) backscattering coefficient, (c1, c2) extinction coefficient, (d1, d2) lidar ratio and (e1, e2) attenuated color ratio.**

Figure 8 (a1) to (e1) present the retrieval results with high data quality during nighttime. In the Eastern European urban agglomeration (area <I>), the total depolarization ratios are 0.08±0.02 and lidar ratios at 532 nm are 50±20 sr. Thus indicating that the urban pollution aerosols may dominate within the boundary layer in this area (Burton et al. 2012). The satellite flew over the Sahara Desert with the footprints cross area <II>. The dominant aerosol type changes from urban pollution aerosols to mixed dust with total depolarization ratio of 0.32±0.03, aerosol lidar ratio of 39±12 sr, and higher extinction coefficient than that within area <I>. In area <III>, some clouds that existed at the range of 6 km and 16 km can be observed over the Atlantic Ocean in the Southern Hemisphere. There are structural features of the aerosol layers in the vertical and horizontal directions as seen in the optical properties of each area, which are often assumed to be a single type aerosol with the same optical characteristics in the absence of HSRL detection capabilities. As shown in area <V> of Fig. 8 (b1), a thin aerosol layer exists at altitudes of 22-26 km, mainly confined between 15°N and 28°S. This layer of aerosols is likely to be smoke or sulfate from the January 2022 Tonga eruption (Legras et al. 2022). Due to the low signal strength of this layer, observations are only available in the 532 nm aerosol backscatter coefficient with the current version algorithm.

The daytime observations of aerosols and clouds by ACDL are influenced by the solar light thus behave lower quality compared to the measurement data during nighttime, as shown in Fig. 8 (a2) to (e2), which are from a segment of data that crosses from the Indian Ocean to the Arabian Peninsula. The ACDL receives a stronger signal from solar background radiation during daylight hours, causing saturation of the detector and faster attenuation of the effective signal when cloud cover is encountered. As shown in area <IV>, where the aerosol layers below the clouds cannot be distinguished from the noise.

The algorithms for processing and retrieval are appropriate for batch operations of ACDL data under various conditions, including daytime and nighttime, different features, and different area. The data production processing rate can meet the satellite data processing needs after proper optimization. As shown in Fig. 9 (a), the total column AOD (Pan et al. 2022) is calculated with the aerosol optical parameters retrieved from ACDL-A data from 1$^{st}$ June to 4$^{th}$ August, 2022. The aerosol profile data are quality screened, most clouds are removed by setting thresholds for the optical properties and then aggregated onto a global 0.25°×0.25° latitude-longitude grid. The global AOD describes the attenuation of light by aerosols and is an important indicator reflecting the degree of air pollution (Levy et al. 2013), The AOD at different heights can also be obtained with ACDL-A. In Fig. 9 (b), the slice diagrams show the global distribution of the layer average AOD in the three height ranges of surface-1 km, 1-3 km, and 3-5 km, respectively.

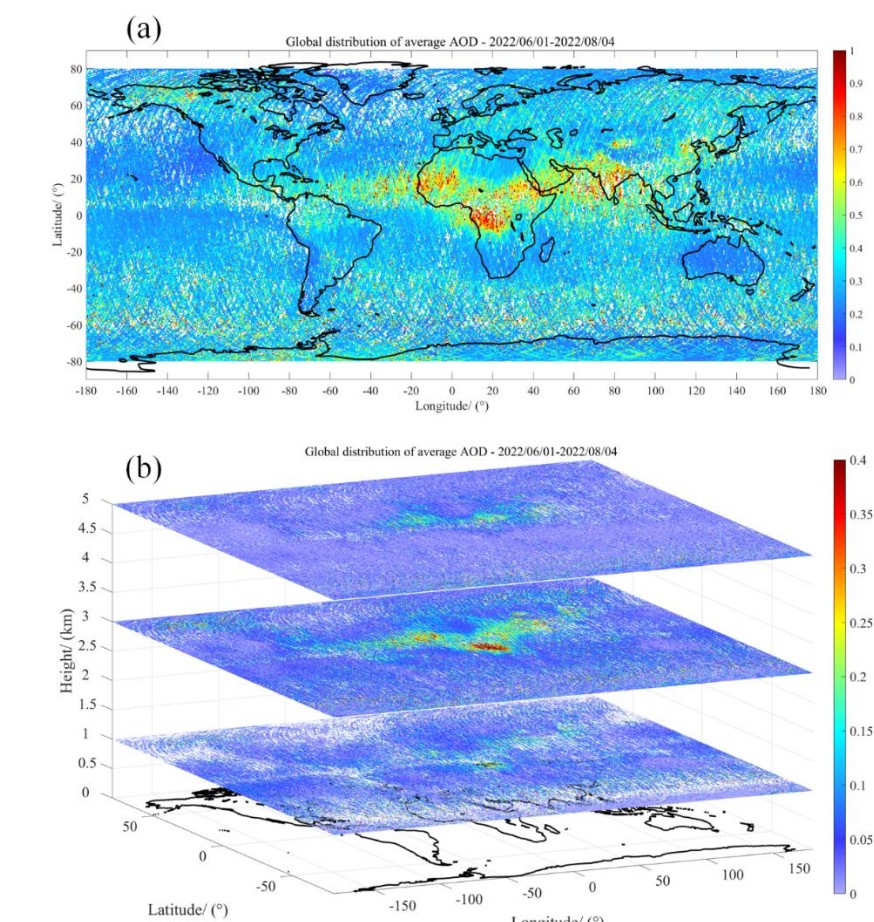

**Figure 9: (a) Global averaged AOD for June 1, 2022 to August 4, 2022. (b) Slices of the global distribution of AOD at different height range from June 1, 2022 to August 4, 2022.**

## 6. Summary and outlook

This paper gives an overview of the main algorithms applied to derive the aerosols and clouds optical properties product of ACDL/DQ-1. ACDL brings a major advance in spaceborne active remote sensing of aerosols and clouds. The ACDL-A data processing algorithms have unique aspects designed to take advantage of these new capabilities. The ACDL-A data processing procedure consists of data preparation, pre-processing and optical properties retrieval. Based on the unique dual-pulse emission configuration and averaging strategy, the algorithm can deliver data products with the horizontal resolution of 3.3 km and vertical resolution of 50 m. Pre-processing algorithms have been developed for the data characteristics of each channel, and data quality control schemes have been designed for both day and night scenarios, after background signals have been removed.. In view of the data characteristics of different channels, a set of sub-channels denoising algorithms are designed specifically for each channel. The capacity of the ACDL-A to independently measure backscatter coefficient and extinction coefficient is

also demonstrated by two observation cases on 27[th] June 2022, where the retrieved lidar ratio is calculated as well. The data processing and retrieval algorithms are applied to the long-term ACDL/DQ-1 observation campaign and a measurement case of the average AOD global distribution is provided.

The ACDL scientific team is gradually accumulating the datasets of optical products as observations in orbit proceed, and related validation activities are ongoing. Although the retrieval of the aerosol optical properties can be realized stably, the

algorithms are still needed to be further optimized. The experience gained from analyzing the data acquired by ACDL has led to many improvements, with significant room for optimization on low quality data and noise control. Various of further algorithm improvement efforts are ongoing. Additionally, the ACDL scientific team is developing a unique scene classification algorithm based on the aerosols and clouds optics products, which requires additional improvement of the algorithms for optical properties retrieval.

**Data availability**

ACDL-A data we used in this paper are not available publicly at the time when the article was submitted. We are allowed to access the data through our participation as a part of ACDL scientific team. The ERA5 dataset are downloaded via the website https://cds.climate.copernicus.eu/cdsapp#!/dataset/reanalysis-era5-pressure-levels?tab=form (last access: 11 July 2023).

**Author contributions**

G. Dai, S. Wu, J. Liu and W. Chen conceived of the idea for the retrieval of the aerosol and cloud optical properties; G. Dai and W. Long wrote the manuscript; G. Dai, S. Wu, W. Long, K. Sun and F. Meng contributed to the algorithm development and data analyses; J. Liu, Y. Xie, Z. Huang and W. Chen contributed to the scientific discussion. All the co-authors reviewed and edited the manuscript.

**Competing interests**

The authors declare that they have no conflict of interest.

**Acknowledgments**

This study has been jointly supported by the Laoshan Laboratory Science and Technology Innovation Projects under grant LSKJ202201202, the National Natural Science Foundation of China (NSFC) under grant U2106210, 61975191 and 41905022.

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
