# Peer review of "Aerosols and Clouds data processing and optical properties retrieval algorithms for the spaceborne ACDL/DQ-1"

_EGUsphere, 2023_

## Referee Comment (RC1)

**Aerosols and Clouds data processing and optical properties retrieval algorithms for the spaceborne ACDL/DQ-1**

by Guangyao Dai et al.

The paper discusses the data and retrieval results of the Aerosol and Carbon Detection Lidar (ACDL) on board the new Chinese DQ-1 satellite launched in April 2022. The ACDL is the first HSRL lidar using an iodine filter in space and measures in 3 wavelengths (532nm, 1064nm and 1572 nm). This work focuses on the retrievals from the first 2 wavelengths.

These exciting first results show some extremely nice images of the different retrieval products (extinction, depolarization, lidar ratio & color-ratio) and promises a great continuation, as well as providing a bridge function, for the current lidar data series starting with CALIPSO & Aeolus missions and the future EarthCARE, AOS and Aeolus-2 missions towards a lidar climate series. To enable the ACDL to take this international role it is important that the L1 data and its retrievals will become available to the general scientific community. We understand that this is not up to the authors, but we do urge that at least the L1 data and L2 products as shown in Figure 8 (June 27 2022; 00:45-01:11 and 09:49-10:15) are provided as part of the 'Data availability' .

There are a number of minor and more important revisions (both technical and textual) which will have to be made before the paper can be accepted for publication. The issues are specified by page and line numbers following the online PDF.

**Revisions:**

In a number of cases too few details are provided when discussing the details of the retrieval algorithms used. I have separated these from the minor textual issues as these require additions to the current text.

**Background noise subtraction (Section 2.3):**

In line 155 it is described that the minimum value of the segmented-averaged signals in the channels is used for subtracting background noise. First to be sure, when you use segment here is it the vertical segment as suggested in Lines 223-229 or do you refer to segments like in Section 5: 'two segments for subsequent data processing'. From the text it is vertical but it is unclear to me how you segment this vertically and if it is the minimum after performing a convolution with a vertical smoothing kernel or it is really the minimum value of any point. Please provide more information on the procedure followed, in case of the first the width of the Kernel, in the second on why this would not provide a very noise behavior

**Auxiliary datasets & denoising**

Line 176: do you have a reference for the C & K system parameters, or is there no additional information available.

In lines 114-116 it states : transmittance of iodine filter for aerosol scattering and molecular scattering are denoted by $fa$ and $fm$, which are function of height due to its dependence on atmospheric temperature and pressure. Later you describe the use of ERA54 which has a relatively low resolution in space and time. Can you provide a discussion of what the errors

are on the retrievals due to the use of the data with respect to higher resolution NWP forecast data or reanalysis. Or is the filter position not that depending on the exact P & T profiles.

In section 4.1.2 (see also Fig. 6) the DNR is an important parameter for quality control. However in the end we only see the profiles in Signal (W [without units]). It would be nice, especially for the molecular channel to also see the corresponding profile of DNR^M with on top the lines you use as threshold for QC.

**Minor revisions:**

There are a number of textual changes to be made and a few issues with the figures. If sentence starts with → a word or ',' should be added

Line 14 add , after 2022

Line 17 → The methods have been

Line 18 → ACDL system and are

Line 21 → ACDL/DQ-1,

Line 24 analysed, which demonstrated → analyzed, demonstrating

Line 29 and absorbing thermal → and absorbing & emitting thermal radiation

Line 33 → active remote sensing tool, lidars can provide aerosol and cloud profile

Line 36 → payload of the Cloud

Line 39-42 → with CALIOP as the  Mie scattering and Rayleigh scattering are combined in the backscatter  signal (Sayer et al. 2012). The CALIOP team has developed the Hybrid Extinction Retrieval Algorithm (HERA) which retrieves both the particulate backscatter and extinction profiles from attenuated backscatter profile by including the scene classification () as a-priori.

Line 43 previous studies : do you point here to both the Young paper or even more, a bit unclear

Line 46 different spectral  → difference in spectral

Line 48 → coefficients can be obtained simultaneously

Line 50 → The Aeolus satellite, which carries a Fabry-Pérot interferometer based wind lidar (ALADIN), was successfully

Line 52 measure → measuring

Line 53 And Aeolus → Additionally Aeolus optimizes the … by using a maximum

Line 56 Add also newer EarthCARE references from the new AMT special issue

Line 61 → DQ-1 is equipped with five sensors,

Line 64 One is → The first is     ; and another is → and the second is

Line 84 → pulses down

Line 88 by the energy → by energy

Line 89 → , which is called dual-pulse

Line 90 → Because of the dual-pulse design,

Line 94  Lidar specialists will know but add something like narrow peak with Mie scattering and broader molecular Rayleigh scattering for the non specialists

Line 150  that only → than only

Line 152 could be higher → can be higher

Line 157 → noise in the high

Line 157-170 Can you give some reference to the values used, e.g. is 1,5 V cross-polar as much as ice clouds would give, or do you only loose dust-aersols??   In Figure 4 the daytime shows consistently peaks at the same place for all 532nm channels. Please provide some context for the reader, i.e. type of clouds at different area's resulting in these noise patterns. It is ofcourse also related to the request above in the revisions requested on the background noise subtraction.

Line 175 Can you provide a reference for the C and K parameters?

Line 186 → raw data in each

Line 187 → scales, a vertical

Line 194 What do you mean with considering the systematic parameters?

Line 200 disorderly noise. Is there something specific you want to say here? Do you mean random noise with respect to bias?

Line 212 Please rephrase first sentences upto distinctly. I don't understand exactly what you want to say

 Line 214 reserve → preserve

Line 219 → different thresholds are chosen.

How is this chosen, are they fixed, do they depend on SNR and are therefore dynamic. Please elaborate on this more.

Line 228  → 2-D median filter, and subsequently a vertical sliding

Please discuss the size of the 2D filter in #profiles and # vertical bins and the size of the vertical sliding window you use for this.

Line 256. On the multiple scattering you refer to Hu & Garnier. But which of the solutions did you pick exactly for ice-clouds and water clouds. And just to be sure, you do neglect multiple scattering by large aerosol particles?

 Line 280 → Thus indicating

Line 281 → satellite passed

Line 284 low-altitude clouds that existed in the range of 6 km and 16km. These are generally not considered low-altitude. Please rephrase

Line 286 in the absence of HSRL. The sentence feels a bit off, maybe add 'HSRLcapabilities' or 'in the case of a backscatter lidar'.

Line 292-295 Move the discussion of area V to the description above describing nighttime

Line 290 Please rephrase sentence 'The superposition …..daytime', it is hard to read and rename echo to backscatter.

Line 296 What do you mean by reusability in this context

Line 297 Can you provide an estimate of the current processing rate with the statement. Every orbit of yy  minutes takes zz minutes to process up to this product.

Line 300 → Most clouds are removed

Line 302 → different heights can also

Line 303 → of the layer averaged AOD

Line 314. Please rephrase sentence 'And the …night scenes'.

Line 325 which require additional improvements

**Figures:**

Figure 2: In the lower box : Average strategy → Averaging strategy

Figure 3: y-axis label Signal/ (V)→ Signal (V) , similar for the x-axis label

Figure 4: You have 3e4 profiles can you provide in the caption how much this is in total time (just to get some reference).

Figure 5: The images look really nice but have a too low resolution (zooming in the label blurs). Can you make the font size slightly larger as well?

Figure 6: Please add units to Signal ^& km in brackets and see comment above on additional plot

Figure 8: The images look really nice but have a too low resolution (zooming in the label blurs). Can you make the font size slightly larger as well? In the caption remove the aerosol 3x in the last sentence. You retrieve both cloud and aerosol properties.

Figure 9: remove whole layer, AOD is the entire range unless specified as layer integrated. Nice way to show the layers btw!!

---

## Author Comment (AC1)

**Aerosols and Clouds data processing and optical properties retrieval algorithms for the spaceborne ACDL/DQ-1**

by Guangyao Dai et al.

The paper discusses the data and retrieval results of the Aerosol and Carbon Detection Lidar (ACDL) on board the new Chinese DQ-1 satellite launched in April 2022. The ACDL is the first HSRL lidar using an iodine filter in space and measures in 3 wavelengths (532nm,1064nm and 1572 nm). This work focuses on the retrievals from the first 2 wavelengths.

These exciting first results show some extremely nice images of the different retrieval products (extinction, depolarization, lidar ratio & color-ratio) and promises a great continuation, as well as providing a bridge function, for the current lidar data series starting with CALIPSO & Aeolus missions and the future EarthCARE, AOS and Aeolus-2 missions towards a lidar climate series. To enable the ACDL to take this international role it is important that the L1 data and its retrievals will become available to the general scientific community. We understand that this is not up to the authors, but we do urge that at least the L1 data and L2 products as shown in Figure 8 (June 27 2022; 00:45-01:11 and 09:49-10:15) are provided as part of the 'Data availability'.

AR: The ACDL science team has been granted permission to upload two data segments related to this post. Each file contains four channels of Level 1 data (stored as "Basic") and Level 2 data products (stored as "Products").

There are a number of minor and more important revisions (both technical and textual) which will have to be made before the paper can be accepted for publication. The issues are specified by page and line numbers following the online PDF.

**Revisions:**

In a number of cases too few details are provided when discussing the details of the retrieval algorithms used. I have separated these from the minor textual issues as these require additions to the current text.

**Background noise subtraction (Section 2.3):**

In line 155 it is described that the minimum value of the segmented-averaged signals in the channels is used for subtracting background noise. First to be sure, when you use segment here is it the vertical segment as suggested in Lines 223-229 or do you refer to segments like in Section 5: 'two segments for subsequent data processing'. From the text it is vertical but it is unclear to me how you segment this vertically and if it is the minimum after performing a convolution with a vertical smoothing kernel or it is really the minimum value of any point. Please provide more information on the procedure followed, in case of the first the width of the Kernel, in the second on why this would not provide a very noise behavior

AR: Section "**3.2 Sub-channel background denoising**" vertically segments the odd-pulse signal profile for each channel to more accurately extract background signals such as solar radiation and system noise. The previous use of 'background noise' may have caused ambiguity, so it has been replaced with the more appropriate 'background signal'. To obtain background signal values that are free of aerosol signals, each odd pulse was divided into multiple segments from high altitude to the subsurface. The smallest average value of each segment was used as the background signal for this set of dual-pulse signals. This method extracts the background signal using one of ICESat-2's calculation methods (Palm et al., 2021).

In section "**4.1.2 Sub-channel denoising algorithm**", the pre-processed signals from each channel are segmented for denoising. The segmentation is still along the vertical direction, and the main purpose is to process the signals with different data quality on the contour separately. The segments are based on DNR thresholds, surface locations, and attenuated locations. Different smoothing and denoising schemes are used for each data segment.

In **Section 5**, the term "two segments for subsequent data processing" refers to the division of observation data in the satellite data retrieval programmer. After a full orbit of observation data, it is divided into daytime and nighttime views. Each daytime or nighttime observation orbit is then divided into two segments of data, generally a low-latitude orbit and a medium-high-latitude orbit, which are stored in two separate files. To reduce read-write pressure on equipment during subsequent data distribution and processing, file sizes are controlled.

**Auxiliary datasets & denoising**

Line 176: do you have a reference for the C & K system parameters, or is there no additional information available.

AR: ACDL uses system constants obtained from system calibration measurements. The development of radiometric calibration algorithms has been carried out in parallel with the in-orbit testing phase. Papers related to the calibration algorithms are being prepared for submission to AMT. The calibration coefficient, C, significantly impacts the accuracy of ACDL data products. The team is working on iteratively updating the calibration scheme through global data validation.

In lines 114-116 it states : transmittance of iodine filter for aerosol scattering and molecular scattering are denoted by fa and fm, which are function of height due to its dependence on atmospheric temperature and pressure. Later you describe the use of ERA54 which has a relatively low resolution in space and time. Can you provide a discussion of what the errors are on the retrievals due to the use of the data with respect to higher resolution NWP forecast data or reanalysis. Or is the filter position not that depending on the exact P & T profiles.

AR: When using ERA5 data, we matched the nearest data horizontally based on latitude and longitude. And then interpolated the data from the 37 pressure gradients to the vertical height at which the ACDL data were located. To ensure processing speed, we established a temperature-pressure- $f_m$ look-up table with 100 Pa pressure resolution and 0.1 K temperature resolution after evaluating the effect of temperature-pressure resolution on data processing.

In section 4.1.2 (see also Fig. 6) the DNR is an important parameter for quality control. However, in the end we only see the profiles in Signal (W [without units]). It would be nice, especially for the molecular channel to also see the corresponding profile of DNR^M with on top the lines you use as threshold for QC.

AR: Thanks for the advice. Corresponding axes for DNR values have been added to the Fig 7.

[Figure]

Figure 7: 532 nm parallel polarized channel (a) and molecular channel (b) denoised profiles, the light color line is the original signal, the dark color line is after denoising.

**Minor revisions:**

There are a number of textual changes to be made and a few issues with the figures. If sentence starts with →a word or ',' should be added

Line 14 add , after 2022

AR: Thanks, revised.

Line 17 →The methods have been

AR: Thanks, revised.

Line 18 →ACDL system and are

AR: Thanks, revised.

Line 21 →ACDL/DQ-1,

AR: Thanks, revised.

Line 24 analysed, which demonstrated →analyzed, demonstrating

AR: Thanks, revised.

Line 29 and absorbing thermal →and absorbing & emitting thermal radiation

AR: Thanks, revised.

Line 33 →active remote sensing tool, lidars can provide aerosol and cloud profile

AR: Thanks, revised.

Line 36 →payload of the Cloud

AR: Thanks, revised.

Line 39-42 →with CALIOP as the Mie scattering and Rayleigh scattering are combined in the backscatter signal (Sayer et al. 2012). The CALIOP team has developed the Hybrid Extinction Retrieval Algorithm (HERA) which retrieves both the particulate backscatter and extinction profiles from attenuated backscatter profile by including the scene classification () as a-priori.

AR: Thanks, revised.

Line 43 previous studies: do you point here to both the Young paper or even more, a bit unclear

AR: Thanks for the advice. The expression has been changed to "The previous validation studies have shown that a relatively large uncertainty would appear in extinction coefficient retrievals of aerosols and clouds as the lidar ratio is selected or modelled (Schuster et al. 2012; Balmes et al. 2019)."

Line 46 different spectral →difference in spectral

AR: Thanks, revised.

Line 48 →coefficients can be obtained simultaneously

AR: Thanks, revised.

Line 50 →The Aeolus satellite, which carries a Fabry-Pérot interferometer based wind lidar (ALADIN), was successfully

AR: Thanks, revised.

Line 52 measure →measuring

AR: Thanks, revised.

Line 53 And Aeolus →Additionally Aeolus optimizes the … by using a maximum

AR: Thanks, revised.

Line 56 Add also newer EarthCARE references from the new AMT special issue

AR: Thanks for your kind remind. The following references have been added:

*Eisinger, M., Marnas, F., Wallace, K., Kubota, T., Tomiyama, N., Ohno, Y., Tanaka, T., Tomita, E., Wehr, T., and Bernaerts, D.: The EarthCARE Mission: Science Data Processing Chain Overview, EGUsphere, 2023, 1-35, 10.5194/egusphere-2023-1998, 2023.*

*Wehr, T., Kubota, T., Tzeremes, G., Wallace, K., Nakatsuka, H., Ohno, Y., Koopman, R., Rusli, S., Kikuchi, M., Eisinger, M., Tanaka, T., Taga, M., Deghaye, P., Tomita, E., and Bernaerts, D.: The EarthCARE mission – science and system overview, Atmos Meas Tech, 16, 3581-3608, 10.5194/amt-16-3581-2023, 2023.*

*Wandinger, U., Haarig, M., Baars, H., Donovan, D., and van Zadelhoff, G.J.: Cloud top heights and aerosol layer properties from EarthCARE lidar observations: the A-CTH and A-ALD products, Atmos Meas Tech, 16, 4031-4052, 10.5194/amt-16-4031-2023, 2023.*

Line 61 →DQ-1 is equipped with five sensors,

AR: Thanks, revised.

Line 64 One is →The first is          ; and another is →and the second is

AR: Thanks, revised.

Line 84 →pulses down

AR: Thanks, revised.

Line 88 by the energy →by energy

AR: Thanks, revised.

Line 89 →, which is called dual-pulse

AR: Thanks, revised.

Line 90 →Because of the dual-pulse design,

AR: Thanks, revised.

Line 94    Lidar specialists will know but add something like narrow peak with Mie scattering and broader molecular Rayleigh scattering for the non specialists

AR: Thanks, the sentence has been revised as "The ACDL takes advantage of the low transmittance valley of the iodine vapor absorption filter at 1110 line to block Mie scattering with a narrow spread. Meanwhile, the broader Rayleigh scattering can pass through (Liu et al., 1997)."

*Reference:*

*Liu, Z.S., Chen, W.B., Zhang, T.L., Hair, J.W., and She, C.Y.: An incoherent Doppler lidar for ground-based atmospheric wind profiling, Applied Physics B, 64, 561-566, 10.1007/s003400050215, 1997.*

Line 150    that only →than only

AR: Thanks, revised.

Line 152 could be higher →can be higher

AR: Thanks, revised.

Line 157 →noise in the high

AR: Thanks, revised.

Line 157-170 Can you give some reference to the values used, e.g. is 1,5 V cross-polar as much as ice clouds would give, or do you only loose dust-aersols??    In Figure 4 the daytime shows consistently peaks at the same place for all 532nm channels. Please provide some context for the reader, i.e. type of clouds at different area's resulting in these noise

patterns. It is of course also related to the request above in the revisions requested on the background noise subtraction.

AR: "The fluctuations in the value of the daytime background signal are related to the intensity of solar radiation at different latitudes, solar energetic events, feature type, and cloud albedo." Figure 4 (a5, b5) has been updated to include the corresponding satellite orbits, and the nighttime and daytime orbits selected in this section are the same as those shown in Figures 5 and 8.

Line 175 Can you provide a reference for the C and K parameters?

AR: The system constant K considers various factors such as detector gain, detector sensitivity, overlap factor, optical efficiency, and so on. The table below provides the approximate values of the parameters for each channel.

| Data channel | K | C(June, 2022) |
|---|---|---|
| 532 nm parallel polarized channel | ~2.9708 e17 | ~0.9±0.2 |
| 532 nm cross-polarized channel | ~8.0921 e17 | ~1±0.2 |
| high-spectral-resolution channel | ~3.7267 e17 | ~0.7±0.1 |
| 1064 nm channel | ~2.2931 e16 | ~1±0.2 |

Papers related to the calibration algorithms are being prepared for submission to AMT.

Line 186 →raw data in each

AR: Thanks, revised.

Line 187 →scales, a vertical

AR: Thanks, revised.

Line 194 What do you mean with considering the systematic parameters?

AR: Thanks, the sentence has been revised as "Figure 5 shows that after implementing the averaging strategy described above, the optical power signals for each channel are calculated using the system parameters and monthly calibration coefficients."

Line 200 disorderly noise. Is there something specific you want to say here? Do you mean random noise with respect to bias?

AR: Thanks, the sentence has been revised as "The molecular channel has an order of magnitude lower signal intensity than the other channels, making it more susceptible to noise, as shown in the Fig 5 (c), which shows the presence of scattering noise."

Line 212 Please rephrase first sentences upto distinctly. I don't understand exactly what you want to say

AR: Thanks for the advice. The sentence has been revised as "After completing the signal averaging process described in the previous sections, the noise still prevents the inversion of the optical parameters for the HSRL method."

Line 214 reserve →preserve

AR: Thanks, revised.

Line 219 →different thresholds are chosen.

AR: Thanks, revised.

How is this chosen, are they fixed, do they depend on SNR and are therefore dynamic. Please elaborate on this more.

AR: Thanks for the advice.

Line 228    →2-D median filter, and subsequently a vertical sliding. Please discuss the size of the 2D filter in #profiles and # vertical bins and the size of the vertical sliding window you use for this.

AR: Thanks for the advice. "To ensure spatial continuity of the observed profiles, the molecular channel applies a 2-D median filter to the signals of the entire orbit using a 5 × 3 window (250m vertically × 6.6km horizontally)."

Line 256. On the multiple scattering you refer to Hu & Garnier. But which of the solutions did you pick exactly for ice-clouds and water clouds. And just to be sure, you do neglect multiple scattering by large aerosol particles?

AR: The effect of multiple scattering is considered in the optical parameter inversion algorithm. As the current scene classification algorithm and cloud phase state classification algorithm are not perfect, a constant value of 0.6 is currently being used for multiple scattering correction. This will remain an ongoing process in future work.

Line 280 →Thus indicating

AR: Thanks, revised.

Line 281 →satellite passed

AR: Thanks, revised as "The satellite flew flied over the Sahara Desert ······"

Line 284 low-altitude clouds that existed in the range of 6 km and 16km. These are generally not considered low-altitude. Please rephrase

AR: Thanks for the advice.

Line 286 in the absence of HSRL. The sentence feels a bit off, maybe add 'HSRLcapabilities' or 'in the case of a backscatter lidar' .

AR: Thanks, revised as "······which are often assumed to be a single type aerosol with the same optical characteristics in the absence of HSRL detection capabilities."

Line 292-295 Move the discussion of area V to the description above describing nighttime

AR: Thanks, revised.

Line 290 Please rephrase sentence 'The superposition …..daytime', it is hard to read and rename echo to backscatter.

AR: Thanks for the advice. The sentence has been revised as "The ACDL receives a stronger signal from solar background radiation during daylight hours, causing saturation of the detector and faster attenuation of the effective signal when cloud cover is encountered."

Line 296 What do you mean by reusability in this context

AR: Thanks for the advice. The sentence has been revised as "The algorithms for processing and retrieval are appropriate for batch operations of ACDL data under various conditions, including daytime and nighttime, different features, and different area."

Line 297 Can you provide an estimate of the current processing rate with the statement. Every orbit of yy   minutes takes zz minutes to process up to this product.

AR: On a home desktop computer with a 2.9GHz CPU, it takes approximately 30 minutes to process and store 25 minutes of orbit data. However, when processing satellite data, multiple orbital data are processed simultaneously using high performance computing servers depending on the task required. And depending on the availability of computing resources, data from different ACDL observation dates are processed simultaneously.

Line 300 →Most clouds are removed

AR: Thanks, revised.

Line 302 →different heights can also

AR: Thanks, revised.

Line 303 →of the layer averaged AOD

AR: Thanks, revised.

Line 314. Please rephrase sentence 'And the …night scenes' .

AR: Thanks for the advice. The sentence has been revised as "Pre-processing algorithms have been developed for the data characteristics of each channel, and data quality control schemes have been designed for both day and night scenarios, after background signals have been removed."

Line 325 →which require additional improvements

AR: Thanks, revised.

**Figures:**

Figure 2: In the lower box : Average strategy →Averaging strategy

AR: Thanks, revised.

[Figure]

Figure 2: Flowchart of the ACDL-A data process. The green box represents the input data, the purple box shows the data processing step, and the blue part indicates the output data products.

Figure 3: y-axis label Signal/ (V)    →Signal (V) , similar for the x-axis label

AR: Thanks, revised.

[Figure]

**Figure 3: Timing diagram for dual-pulse emission and acquisition.**

Figure 4: You have 3e4 profiles can you provide in the caption how much this is in total time (just to get some reference).

AR: Thanks, revised.

[Figure]

**Figure 4: The background signal value subtracted from each echo signal in nighttime (blue dash on the left) or daytime (orange dash on the right) segment of data, both on June 27, 2022. Corresponding to each channel in order from top to bottom, (a1, b1) parallel-polarized channel at 532 nm, (a2, b2) cross-polarized channel at 532 nm, (a3, b3) molecular channel at 532 nm, (a4, b4) 1064 nm channel, and (a5, b5) satellite orbit.**

Figure 5: The images look really nice but have a too low resolution (zooming in the label blurs). Can you make the font size slightly larger as well?

AR: Thanks, revised.

[Figure]

**Figure 5: ACDL-A optical power signals in (a) 532 nm parallel polarized channel, (b) 532 nm perpendicular polarized channel, (c) 532 nm molecular channel and (d) 1064 nm channel on 00:45:23 UTC to 01:10:22 UTC on June 27, 2022.**

Figure 7: Please add units to Signal ^&   km in brackets and see comment above on additional plot

AR: Thanks, revised.

[Figure]

**Figure 7: 532 nm parallel polarized channel (a) and molecular channel (b) denoised profiles, the light color line is the original signal, the dark color line is after denoising.**

Figure 8: The images look really nice but have a too low resolution (zooming in the label blurs). Can you make the font size slightly larger as well? In the caption remove the aerosol 3x in the last sentence. You retrieve both cloud and aerosol properties.

AR: Thanks, revised.

[Figure]

**Figure 8: Retrievals from ACDL-A measurements between 00:45:23 and 01:10:22 (nighttime, a1 to e1) and between 09:49:05 and 10:14:05 (daytime, a2 to e2) on June 27, 2022. From top to bottom: The time series of (a1, a2) total depolarization ratio, (b1, b2) backscattering coefficient, (c1, c2) extinction coefficient, (d1, d2) lidar ratio and (e1, e2) attenuated color ratio.**

Figure 9: remove whole layer, AOD is the entire range unless specified as layer integrated. Nice way to show the layers btw!!

AR: Thanks, revised.

---

## Author Comment (AC2)

**Responses to RC2:**

The Aerosol and Carbon Detection Lidar (ACDL) is the first high spectral resolution lidar using an iodine filter in space. This is an important milestone for aerosol and cloud research from space and thus has the opportunity to advance our understanding of aerosols, clouds and their interaction, once the data will hopefully be made publicly. The manuscript describes the data, retrieval and first results of ACDL with focus on the aerosol (cloud) retrieval (ACDL-A). Thus, the paper is very important with respect to future use of the data, especially if viewed as a piece of documentation of the aerosol retrieval for DQ-1. However, for me to accept the paper, revisions are needed, as too many details of the processing are missing. In the block diagram Fig.6 there are some crucial processing steps like 'Wavelet domain denoising' or 'Multi scale local denoising' which are not described at all. There is also no description of the depolarization calibration.

AR: Thanks for the valuable advices. In section 4.1.2 of the revised manuscript, a more detailed description of the chosen filtering and noise reduction scheme is given.

The ACDL system has undergone ground calibration, including depolarization calibration. The calibration module has been specially reserved in the spaceborne ACDL. In this module, a calibration beam with a known polarization state is pre-set. This, in combination with the usage of a half-wave plate, can be used for the on-orbit polarization calibrations. This method has been well introduced in Alvarez et al., 2006 and Freudenthaler, 2016. The science team is currently evaluating and analyzing the system's performance during on-orbit calibration and preparing for on-orbit polarization calibration.

*Reference:*

*Alvarez, J.M., Vaughan, M.A., Hostetler, C.A., Hunt, W.H., and Winker, D.M.: Calibration Technique for Polarization-Sensitive Lidars, J Atmos Ocean Tech, 23, 683-699, https://doi.org/10.1175/JTECH1872.1, 2006.*

*Freudenthaler, V.: About the effects of polarising optics on lidar signals and the Δ90-calibration, Atmos Meas Tech, 9, 4181-4255, 10.5194/amt-9-4181-2016, 2016.*

Specific comments:

P1, l.16: 'two wavelength polarization detection' gives the impression that the depolarization is also detected for the 1064 nm channel. But according to the block diagram this is not the case.

AR: Thanks for the advice. Sentences with ambiguous expressions have been replaced as "The ACDL/DQ-1 is a high-spectral-resolution lidar (HSRL) that separates molecular backscatter signals using an iodine filter, and has 532nm polarization detection and dual wavelength detection at 532nm and 1064nm, which can be utilized to derive aerosol optical properties."

P2, l.39: Please state once for non-lidar specialists what the lidar ratio is.

AR: Thanks for the advice. The corresponding description "Lidar ratio is defined as the ratio of the aerosol extinction coefficient to the backscattering coefficient and is closely related to the physical and optical properties of the particles." has been added.

P2, l.50: It would be good to also include the first papers that proposed the HSRL technique.

AR: Thanks for the advice. Relevant references have been added to the paper: "Taking advantage of the different spectral broadening, high-spectral-resolution lidar (HSRL) can separate the aerosol contribution from the molecular backscatter with a narrow bandwidth optical filter (Fiocco and DeWolf, 1968; Shimizu et al., 1983). Thus, without assuming the lidar ratio, the aerosol backscatter and extinction coefficients could be obtained respectively. HSRL uses several techniques to achieve a clear separation between Mie and Rayleigh scattering spectra, including the Fabry-Pérot interferometer edge technique approach (Garnier and Chanin, 1992; Flesia and Korb, 1999), interferometric fringe imaging techniques (Matthew and James, 1998) and atomic or molecular filter discrimination (She et al., 1992; Liu et al., 1997)."

*Reference:*

*Fiocco, G., and DeWolf, J.B.: Frequency Spectrum of Laser Echoes from Atmospheric Constituents and Determination of the Aerosol Content of Air, Journal of Atmospheric Sciences, 25, 488-496, https://doi.org/10.1175/1520-0469, 1968.*

*Flesia, C., and Korb, C.L.: Theory of the double-edge molecular technique for Doppler lidar wind measurement, Appl Optics, 38, 432-440, 10.1364/AO.38.000432, 1999.*

*Garnier, A., and Chanin, M.L.: Description of a Doppler rayleigh LIDAR for measuring winds in the middle atmosphere, Applied Physics B, 55, 35-40, 10.1007/BF00348610, 1992.*

*Liu, Z.S., Chen, W.B., Zhang, T.L., Hair, J.W., and She, C.Y.: An incoherent Doppler lidar for ground-based atmospheric wind profiling, Applied Physics B, 64, 561-566, 10.1007/s003400050215, 1997.*

*Matthew, J.M., and James, D.S.: Comparison of two direct-detection Doppler lidar techniques, Opt Eng, 37, 2675-2686, 10.1117/1.601804, 1998.*

*Shimizu, H., Lee, S.A., and She, C.Y.: High spectral resolution lidar system with atomic blocking filters for measuring atmospheric parameters, Appl Optics, 22, 1373-1381, 10.1364/AO.22.001373, 1983.*

*She, C.Y., Alvarez, R.J., Caldwell, L.M., and Krueger, D.A.: High-spectral-resolution Rayleigh–Mie lidar measurement of aerosol and atmospheric profiles, Opt Lett, 17, 541-543, 10.1364/OL.17.000541, 1992.*

P4, Figure 1: Besides the block diagram a table containing basic system parameters (rep. rate, pulse energy, telescope diameter, detector type, sensitivity, …) should be included. Some are mentioned in the text, but it is best to put them together in one place.

AR: Thanks for the advice. A summary table of system parameters has been added to the paper.

Table 1: Parameters of the ACDL instrument

| Parameters | Value |
|---|---|
| Wavelength | 532.024 nm; 1064.490 nm |
| Pulse Energy | ~130 mJ@532 nm; ~180 mJ@1064 nm |
| Laser frequency stability | <2 MHz (RMS) |
| Laser divergence Angle | ≤60 μrad@532/1064 nm; |
| Gain | 59.46@parallel; 53.4573@vertical; 32@HSRL |
| Telescope diameter | 1.0 m |
| Lidar Off-Nadir Angle | 2° |
| Laser Repetition Frequency | 20 Hz @ dual-pulse |
| Sampling rate | 50 MHz |
| Vertical Resolution (raw data) | 3 m@<7.5 km; 24 m (8 bin average) @>7.5 km |
| Horizontal Resolution (raw data) | ~ 330 m |

P5, l.110: If z is height (altitude), this assumes an exactly nadir pointing lidar. Since this is not the case, there are some terms missing to account for off nadir pointing.

AR: Thanks for your kind reminder. For data processing, we calculated the elevation of each bin, taking into account the Lidar Off-Nadir Angle. We also standardized the vertical height information of the data products to orthometric height for user convenience. The description has been added, as "It is to be pointed out that in the data processing work in this paper, all heights are standardised to the orthometric height, where z is the altitude to the local geodetic level."

P5, l.115: $f_a$ should not depend on temperature and pressure, only $f_m$

AR: Thanks for the advice. Sentences with ambiguous expressions have been replaced as "The transmittance of iodine filter for molecular scattering is denoted by $f_m$, which are function of height due to its dependence on atmospheric temperature and pressure. And the transmittance of iodine filter for aerosol scattering is denoted by $f_a$."

P5, l.119: Can you give a reference, please?

AR: Thanks for the advice. Added the following two references for inversion of aerosol optical parameters by coupled equations:

*Hair, J.W., Hostetler, C.A., Cook, A.L., Harper, D.B., Ferrare, R.A., Mack, T.L., Welch, W., Izquierdo, L.R., and Hovis, F.E.: Airborne High Spectral Resolution Lidar for profiling aerosol optical properties, Appl Optics, 47, 6734-6752, 10.1364/AO.47.006734, 2008.*

*Liu, D., Yang, Y., Cheng, Z., Huang, H., Zhang, B., Ling, T., and Shen, Y.: Retrieval and analysis of a polarized high-spectral-resolution lidar for profiling aerosol optical properties, Opt Express, 21, 13084-13093, 10.1364/OE.21.013084, 2013.*

P6, l.1443: Launch = emission?

AR: Thanks for the advice. The word has been replaced with "emission".

P6, l.146: It is not clear, what is meant here. The two pulses are already separated in time.

AR: The double pulse utilizes a distinctive data acquisition method. The return set of Dual-pulse signals consists of the 1st to 4824th bin of the odd-pulse, the signals for the even-pulse begin with the 4825th to 9152h bin.

"Since the original signal contains two pulses with different height resolution data, it is necessary to match odd-pulse and even-pulse in altitude separately in the data preparation phase. Based on the time of emission and acquisition, the position of each data point relative to the satellite can be calculated. The latitude and longitude of the laser footprint points corresponding to each set of dual-pulse were determined using spacecraft attitude and ephemeris data. The ellipsoidal heights corresponding to each data point of the odd-pulse and even-pulse are calculated separately by the WGS-84 (Lohmar 1988) coordinate system, then converted to orthometric height using the geoid height. For certain vertical resolution requirements in subsequent processing, the odd and even pulses will be averaged bin by bin by matching them to the appropriate height interval."

P7, l.155: 'The mean signals' would be better, as a contrast to the 'minimum values' for the other channels. And only the mean offset can be estimated and subtracted and not the total 'noise' as stated in the text.

AR: In certain areas, the maximum detection altitude is only about 35 km, depending on the satellite's orbital altitude and acquisition start time. To obtain background signal values that are free of aerosol signals, each odd pulse was divided into multiple segments from high altitude to the subsurface. The smallest average value of each segment was used as the background signal for this set of dual-pulse signals. This approach was used for all three channels of the 532 nm. The background signal value for 1064nm is determined by averaging the data from the high-altitude segments (average of the 100th~300th acquisition data of the odd-pulse), following conventional practice.

This method extracts the background signal using one of ICESat-2's calculation methods (Palm et al., 2021). It has been successfully applied in batch processing of both daytime and nighttime data.

*Reference:*

*Palm, S.P., Yang, Y., Herzfeld, U., Hancock, D., Hayes, A., Selmer, P., Hart, W., and Hlavka, D.: ICESat-2 Atmospheric Channel Description, Data Processing and First Results, Earth Space Sci, 8, e1470E-e2020E, https://doi.org/10.1029/2020EA001470, 2021.*

P8, l.159: It is not the 'background noise' but the 'background signal'.

AR: Thanks for the advice. All inaccurate expressions in the full paper have been replaced with relevant and precise language.

P9, Figure 5: Font of scale and axis titles are too small

AR: Thanks for the advice, revised.

[Figure]

**Figure 5: ACDL-A optical power signals in (a) 532 nm parallel polarized channel, (b) 532 nm perpendicular polarized channel, (c) 532 nm molecular channel and (d) 1064 nm channel on 00:45:23 UTC to 01:10:22 UTC on June 27, 2022.**

P10, l.228: Median filters are not linear and do not preserve mean values. How large is the window for this? What is the size of the sliding window

AR: The molecular channel profiles along the orbitals exhibit comparable salt-and-pepper noise, which could negatively affect the subsequent optical parameter inversion. To address this issue, the median filter proves to be effective in eliminating the salt-and-pepper noise. "The molecular channel applies a median filter to the signals of the entire orbit using a 5 ×

3 window (250m vertically × 6.6km horizontally)." And using a 20-point sliding window to extracts low-quality signals in the high-altitude, subsurface, and totally attenuated regions. "The sliding window method ensures that occasional spikes in the signal do not interrupt the continuity of data segmentation."

P11, Equ.8: What numerical scheme is used to calculate the derivative? And in calculating the lidar-ratio, what measures are taken that alpha and beta have the same vertical resolution?

AR: The Equation $\frac{\tau(z)-\tau(z-1)}{\Delta z}$ is used to calculate the total extinction at height z. $\tau$ (z) is the optical depth profile and $\Delta z$ is the vertical resolution of the profile, which is 50m. This paper presents the aerosols and clouds optical properties products of the ACDL/DQ-1, all of which have a uniform resolution of 50 m in the vertical and 3.3 km in the horizontal.

p.12, l.249: This paper gives only the basic algorithm. What values for bulk- and sheer viscosity and thermal conductivity and their temperature dependence are used? Please give a reference!

AR: Thanks for the advice. Added the following two references

*Shneider, M.N., Miles, R.B., and Pan, X.: Coherent Rayleigh-Brillouin scattering in molecular gases, Phys Rev a, 69, 33814, 10.1103/PhysRevA.69.033814, 2004.*

*Gu, Z., Witschas, B., van de Water, W., and Ubachs, W.: Rayleigh–Brillouin scattering profiles of air at different temperatures and pressures, Appl Optics, 52, 4640-4651, 10.1364/AO.52.004640, 2013.*

p.13, Figure 8: Fonts are too small.

AR: Thanks for the advice, revised.

[Figure]

**Figure 8:** Retrievals from ACDL-A measurements between 00:45:23 and 01:10:22 (nighttime, a1 to e1) and between 09:49:05 and 10:14:05 (daytime, a2 to e2) on June 27, 2022. From top to bottom: The time series of

**(a1, a2) total depolarization ratio, (b1, b2) backscattering coefficient, (c1, c2) extinction coefficient, (d1, d2) lidar ratio and (e1, e2) attenuated color ratio.**

P14, l281: flied = flew?

AR: Thanks for the advice. The word has been replaced with more commonly used phrases.

---

## Author Response (AR2)

**Responses to RC:**

Line 52: the different in --> the difference in

Line 252: sym4 wavelet may require an additional reference for those not familiar with using wavelets

Line 258: reorder the sentence 'the molecular channel applies a 2-D median filter to the signals' to 'a 2-D median filter is applied to the molecular channel signals'

AR: We sincerely thank the editor and all reviewers for their valuable feedback that we have used to improve the quality of our manuscript. Grammatical expressions have been revised, and added the following two references:

[revised manuscript text omitted]